# Supramarginal Resection for Glioblastoma: It Is Time to Set Boundaries! A Critical Review on a Hot Topic

**DOI:** 10.3390/brainsci12050652

**Published:** 2022-05-16

**Authors:** Francesco Guerrini, Elena Roca, Giannantonio Spena

**Affiliations:** 1Unit of Neurosurgery, Department of Surgical Sciences, Hospital Santa Maria Goretti, 04100 Latina, Italy; 2Head and Neck Department, Neurosurgery, Istituto Ospedaliero Fondazione Poliambulanza, 25124 Brescia, Italy; rocaelena@gmail.com; 3Technology for Health PhD Program, University of Brescia, 25124 Brescia, Italy; 4IRCCS Fondazione Policlinico San Matteo, 27100 Pavia, Italy; giannantonios@gmail.com

**Keywords:** Glioblastoma, high-grade glioma, supramarginal resection, Flairectomy

## Abstract

Glioblastoma are the most common primary malignant brain tumors with a highly infiltrative behavior. The extent of resection of the enhancing component has been shown to be correlated to survival. Recently, it has been proposed to move the resection beyond the contrast-enhanced portion into the MR hyper intense tissue which typically surrounds the tumor, the so-called supra marginal resection (SMR). Though it should be associated with better overall survival (OS), a potential harmful resection must be avoided in order not to create new neurological deficits. Through this work, we aimed to perform a critical review of SMR in patients with Glioblastoma. A Medline database search and a pooled meta-analysis of HRs were conducted; 19 articles were included. Meta-analysis revealed a pooled OS HR of 0.64 (*p* = *0.052*). SMR is generally considered as the resection of any T1w gadolinium-enhanced tumor exceeding FLAIR volume, but no consensus exists about the amount of volume that must be resected to have an OS gain. Equally, the role and the weight of several pre-operative features (tumor volume, location, eloquence, etc.), the intraoperative methods to extend resection, and the post-operative deficits, need to be considered more deeply in future studies.

## 1. Introduction

Glioblastoma represents the most common primary malignant brain tumor in adults with a less than 7% 5-year survival rate despite complete resection and adjuvant therapies. By definition, it is an incurable disease due to its infiltrative nature which invariably and typically brings relapse around the surgical cavity or, less frequently, into new remote tumoral foci [1]. This peculiarity was already noted by Dandy who tried to eliminate most of the tumor by performing a hemispherectomy, with frustrating results in terms of neurological deficit and, ultimately, survival. That attempt represented the first experience with supramarginal resection (SMR) [2].

Usually, Glioblastoma is depicted on MRI as a contrast-enhanced nodule surrounded by a T2 hyperintense signal of variable shape and extension; this hyper signal is generally thought to represent infiltrated brain. Surgery has classically aimed toward the complete resection of the MRI T1w sequence enhancing tumor and accumulated data show the benefit of complete resection over partial resection or biopsy [3,4]. Understandably, several authors investigated if resecting brain matter beyond the contrast-enhanced area could add further improvements in overall survival (OS). Despite the wide range of variability in patients’ cohorts, selection criteria, surgical techniques, and evaluation of results, the majority of researchers agreed with the fact that SMR could improve OS.

Although it could seem logical that, for an infiltrating disease, the more tumor is eliminated the more positive effects can be expected, in Glioblastoma, the matter of maintaining a balance between aggressive resection and neurological functioning is of particular interest due to the short survival and the eventual lack of time for recovery. Still, looking at the available evidence, many issues are unresolved, starting from the definition of SMR itself. Similarly, the extent of resection after which an SMR is obtained and how to assess it is not clear. Are there intraoperative adjuncts that guide the surgeon to push resection beyond the contrast-enhanced edges? More generally, can an SMR be proposed for all Glioblastoma patients? Are there pre-operative selection criteria for SMR? How can SMR be balanced with the functional outcome? What is the relationship between SMR and standard prognostic factors (i.e., age, volume, site, genetic features, involvement of periventricular zone, etc.)?

In this work, we performed a critical literature review on SMR on patients affected by Glioblastoma, trying to shed light on many debated points that need to be elucidated before it can be considered a clearly advantageous treatment.

## 2. Materials and Methods

A search was conducted on Medline Database using the following keywords: “Supramarginal resection”, “Supratotal resection”, “FLAIR resection”, “FLAIRectomy”, and “Lobectomy” associated with “Glioblastoma” OR “Glioma” OR “High grade Glioma”. Relevant papers were also obtained by references sections. A clinical series and literature review were included in this analysis. Articles were excluded if they concerned low-grade gliomas. Case reports, surveys, and consensus statements were excluded from the review, but they were considered for the discussion. Moreover, whenever it was possible, overall survival Hazard Ratios were collected and analyzed in a pooled meta-analysis. Specifically, it was performed using R software (R Core Team, Wien, Austria) and the package “metafor” [5]. Multilevel linear mixed effects models with an unstructured covariance matrix were used for the meta-analysis in order to account for the clustering of data deriving from the same studies. *p* values < 0.05 were considered statistically significant.

## 3. Results

A total of 19 articles were included, with 2 literature reviews [6,7] and 17 clinical series. The majority of them defined SMR as the resection of any FLAIR volume beyond T1 contrast-enhanced tumor edges. Meanwhile, papers by Roh et al., Schneider et al., and Figueroa et al. equated SMR to the concept of lobectomy; particularly, the two last papers showed the results of a standard or a minimally invasive anterior temporal lobectomy [8,9]. Finally, IONM represented the most commonly used intraoperative method.

A paper by Karshnia et al. was excluded because it was a consensus of recommendations [10], while another by Rakovec et al. was not included as it was a survey [11]. The work by Borger et al. was excluded considering that the authors analyzed the seizure outcome of the same patients as Schneider et al., already included in this review [12]. Table 1 shows all the available data extracted from the analyzed papers. Considering that the single-center retrospective observational cohort studies represented the greatest part of the included papers, the level of evidence could be summarized as class C-EO [13].

As far as meta-analysis was concerned, some papers were excluded. A study by Roh et al. was discharged because HR was reported as the ratio between SupTR and GTR [14]; works by Pessina et al. and Shah et al. did not show HR [15,16]; a paper by Certo et al. described a correlation index and was not included [17]; only representation of Kaplan-Meier curves by Altieri et al. and Aldave et al. led to the exclusion of these studies [18,19]; and a paper by Mampre et al. was excluded due to the hazard-ratio being calculated for post-operative volumes and not according to extent of resection [20]. The work of Eyopoglu et al. was excluded as it was not a controlled study [21]. As for a paper by Tripathi et al., HR was considered for every t ype of glioma but, belonging to the same study, they were interpreted as correlated [22]. Finally, since Li et al. described SMR HR for a FLAIR resection of at least 53.21%, only this result was included in this work [23].

Statistical analysis revealed a high and significant heterogeneity between studies (*p* < 0.001). The calculated pooled HR was 0.64 (0.41–1.00, *p* = **0.052**) (Figure 1).

## 4. Discussion

Glioblastoma continues to represent a hard therapeutic challenge considering its high malignancy due to its infiltrative nature. A recent 2021 WHO Classification abutted the term “glioblastoma” to only wild-type IDH gliomas with certain pathological and molecular features; on the contrary, we are allowed to include a “grade 4 astrocytoma” if the IDH gene is mutated. The difference is substantial, considering the clinical course, therapeutic response, and patient prognosis [24,25]. Whatever the definition, one of the mainstays for surgery of gliomas pivots around the so-called “maximal safe resection” which implies balancing oncological useful resection with maintaining optimal neurological functioning. In fact, both of these goals (total removal and neurological integrity) equally influence the prognosis [26].

### 4.1. Definition of SMR, Its Impact on OS, and the Role of FLAIR

There is a discrete consensus about the definition of SMR since the majority of the studies state that SMR is the resection of any part of a T1w gadolinium-enhanced tumor exceeding FLAIR volume.

However, far less agreement can be found concerning the role of SMR and its impact on OS. In fact, in contrast to the well-established role of the EOR of the contrast-enhanced component, when considering the so-called “FLAIR-ectomy” there is much less agreement. Li et al. found that a resection larger than 53.2% of FLAIR volume confers an advantage on OS in previously untreated IDH mutated patients [23]. A quite similar value was found by Pessina et al. and Tripathi et al. who conducted a different analysis according to the radiological appearance of the tumor; while SMR seems to bring an OS advantage in patients with moderately and highly diffuse wtIDH glioblastomas; in case of nodular ones, a maximum of 29% of FLAIR volume would be advantageous [24]. Additionally, Yan et al. found the DTI sequence’s anisotropic component to bepositively associated with OS and PFS [27]. The review by Karschnia et al. tried to clarify the topic concluding that SMR is defined as any resection beyond contrast enhancement into T2w/FLAIR hyperintensity [10].

It is evident that many of the uncertainties come from the real significance of the hyperintense T2 signal. It is generally found that the GBM relapses just beyond the resected contrast-enhanced edges, as some works claimed it is plausible to think that stem-like cells can be found in this area [28]. Studies comparing MR and 18FET-PET demonstrated that the T2 hyper signal is likely to host tumor cells [29]. However, PET imaging is an advanced modality that is not available in every center, so the majority of the studies are based on standard MR imaging. From this perspective, the FLAIR hyper signal has been retained as a marker of tumoral infiltration, although it cannot actually distinguish infiltration from brain edema. To resolve that issue, a FLAIR hyper signal can be found around brain metastasis too, reflecting a vasogenic edema [30]. Some studies tried to develop methods to differentiate tumor infiltration areas from edema. Certo et al. described a manual segmentation method that distinguished Region of Interests (ROIs) with different hyperintensity values on FLAIR sequences; ROIs with higher values represented edema [17]. Other studies revealed that Apparent Diffusion Coefficient (ADC) mapping can have a prognostic value in patients with a Glioblastoma, as it reflects water sequestration and, as a consequence, hypercellularity. Elson et al. found that a <0.3 minimum ADC value was associated with a shorter OS and PFS [31]. Finally, a peritumoral FLAIR hyper signal could have a different significance, and hence, the benefit of removing apparently healthy tissue beyond the contrast-enhancing “meaty” tissue has conflicting evidence.

### 4.2. SMR, Tumor Volume, and Location

When talking about the resection of an infiltrating brain tumor, the first questions that arise in the surgeon’s mind are about the location (eloquent/critical area versus less “dangerous” areas) and volume, which are intimately bound. A large part of the studies included in this review showed a wide range of pre-operative tumor volumes and only a few differentiated between eloquent and non-eloquent locations. For example, Vivaz-Butraigo et al. reported a range between 1 and 124 cm^3^ for contrast-enhanced volumes that reached 182.74 cm^3^ in FLAIR sequences. They reported a positive influence in cases of 20% to 50% SMR, without a clear advantage for greater resection [32].

Although in a standardized predictive model for Glioblastoma pre-operative tumors volume is not usually considered, it is intuitive that larger tumors can hamper SMR, especially when coupled to an eloquent or near-eloquent location. Such a consideration found important feedback in the work by Roh et al. which showed that a frontal or temporal lobectomy for glioblastoma located in the non-dominant hemisphere was associated with longer OS and PFS [14]. Schneider et al. showed an anterior temporal lobectomy was able to prolong both OS and PFS, both on the dominant and non-dominant sides. On the contrary, Figueroa et al. did not find a survival advantage in using this technique. Regardless, according to the survey by Rakovec et al., the neurosurgical oncology community seems to agree with limiting the SMR to right anterior temporal and right frontal lobe GBM [11].

In some cases, such as a small GBM located in the eloquent area, SMR can be advocated (see Figure 2); however, this only concerns single cases and a very attentive selection. Other than these very specific cases, it seems hard to conceive that patients with larger lesions near or inside eloquent areas could benefit from SMR without putting them at higher risk of neurological outcomes. As discussed below, accumulated evidence demonstrated that a worsened post-operative neurological performance abrogates survival benefits from complete tumor resection or unilobed tumor location [33,34,35]. An analog consideration could be done for tumors infiltrating deep neural and vascular structures (i.e., insular glioblastoma) which typically represent the boundaries of the resection cavity. It appears that a deeper analysis is needed in order to clarify the role of tumor volume and location on SMR, with special attention regarding the eloquent area location.

### 4.3. SMR and Tumors Infiltrating Periventricular White Matter

Tumor-initiating brain cells are thought to be placed into the so-called Subventricular Zone (SVZ) and evidence supporting this assumption is still accumulating. The contact between cancerous cells and SVZ seems to confer higher resistance to traditional radio- and chemotherapy [36]. Additionally, a lateral ventricle wall involvement is considered the source of leptomeningeal dissemination and, finally, for obstructive hydrocephalus [37]. Hallaert et al. analyzed 214 patients and found that contact with SVZ was associated with unmethylated MGMT and a shorter OS [38]. Vivaz-Buitrago et al. and Tripathi et al. examined the involvement of the lateral ventricle, confirming its role as a poor prognostic factor in case the contrast-enhancement reaches the ependyma [32]. However, the role of the infiltration of the ependyma by the FLAIR hyper signal received less attention. Mistry et al. demonstrated that the distance between glioblastoma and SVZ did not influence OS, which, on the contrary, suffers from the contact between SVZ and contrast-enhancement edges [22]. This data seems conflicting since as long as the tumor grows toward the SVZ, it is difficult to explain why it appears true for the CE portion only. In other words, if FLAIR volume contains tumoral cells, it should have the same role in the SVZ involvement. As a consequence, does the resection of this volume confer increased OS? Undoubtedly, more studies are necessary and this factor should be well analyzed if the neurosurgical community wants to establish criteria for SMR in the case of high-grade gliomas.

### 4.4. SMR and Intraoperative Techniques

There are several technological tools used intraoperatively to guide tumor resection which help in better visualizing tumor tissue such as intraoperative MRI, ultrasound, fluorescent agents, and 5-ALA. These latter two are specifically addressed to detect tissue that corresponds to the contrast-enhancing MR images.

There is no clarity on how intraoperative technologies make SMR feasible in glioma surgery. Certainly, the use of iMR (intraoperative magnetic resonance) and fluorophores helps in the best extension of tumor resection [39,40].

When coming to the eloquent location of tumors, intraoperative mapping is useful to guide resection according to functional boundaries rather than only anatomical. This strategy has shown robust results both for LGG and HGG. However, despite the use of functional monitoring and mapping, the idea to pursue aggressive resection in GBM has to take into account the fact that rapidly growing tumors bring a more destructive behavior compared to their slow-growing counterparts. This biological difference implies that brain plasticity has much less time to intervene, not allowing the brain to reshape and potentially increasing the risk for post-operative definitive impairments [41].

Pessina et al. performed a resection guided by neuronavigation and ultrasounds, extending until cortical and subcortical stimulation enhanced the risk of neurological deficits [15].

Some authors assert that sodium fluorescein, which accumulates in the extracellular space when the barrier is damaged, may be the intraoperative equivalent of the radiological signal given by gadolinium. The same authors state that this marker extends beyond tumor regions with contrast pinch and therefore can predict the pathological tissue facilitating resection [42]. Other authors recommend associating fluorescein with Raman spectroscopy which has been shown to be able to identify tumor versus healthy tissue at the margins of resection [43]. Furthermore, laser endomicroscopy, associated with fluorescein, can also have the same effect [44,45,46]. However, the use of 5-ALA appears questionable. In fact, despite Eyopoglu et al. finding an OS advantage in the DiVA group, Roh et al. did not obtain a better survival in their subgroup of patients in which 5-ALA was employed [21,32]. Nevertheless, experiences are limited and they cannot be elevated as a standard methodology.

### 4.5. SMR and Functional Outcome

It has been established that the total or near-total resection of the contrast-enhancing component is a strong predictor of prolonged OS [47,48,49,50,51]. In more recent years, it has also been demonstrated how radical resection must be balanced with the preservation of adequate functional outcomes, since this latter can negatively affect the deployment of adjuvant treatments and the OS. A first retrospective study published by McGirt et al. on Glioblastoma patients who received tumor resection introduced the prominent role of surgically acquired language and motor deficit on survival impairment (9.0-months and 9.6-months median survival, respectively, compared to 12.8 months without a new deficit, *p* < 0.05) [33]. Furthermore, in 2015, Verlut et al. showed that post-operative motor deterioration was associated with poor outcomes in patients receiving surgery followed chemo-radiotherapy [46]. Specifically, it has more recently been demonstrated how severe post-operative neurological deficits significantly reduce survival rates and become a predominant negative prognosticator over EOR, tumor location, KPS, age at the date of surgery, MGMT promoter methylation status, and adjuvant treatment regimen. Rahman and colleagues published a comprehensive work demonstrating that post-surgical acquired neurological deficits abrogate the survival benefit gained by EOR of 95% and more [34].

So, if these considerations hold true for radical resection, it is still more of a concern when dealing with SMR, especially when tumors are located near or inside eloquent locations. Aabaedi et al. claimed that as far as the subgroup of wild-type IDH Glioblastoma is concerned, no difference between CE and non-CE resection emerged in terms of OS; on the contrary, they reinforced the concept that a neurological impairment represents the real key factor for survival of these patients [52].

The greatest part of the papers included in our review compared pre- and post-operative KPS and they did not find any difference, concluding that SMR was not associated with a worse clinical outcome. Nevertheless, even if KPS has a great value in the oncological field, it is necessary to remember that it does not explore every clinical aspect. In fact, less attention has been paid to post-operative cognitive status evaluation; only a few studies in this review conducted a deep analysis of specific symptoms, above all neurocognitive ones. As emerged from the review by Gately et al., a longer OS should be balanced with a certain quality of life that permits them to maintain real functional independence [53]. Therefore, as Roh et al. stated, a study that analyzes the effects of SMR on neuropsychological functions is desirable [14].

### 4.6. Pooled HR Meta-Analysis

We conducted a pooled meta-analysis to summarize overall survival HR resulting from every study included as described earlier. The first element that emerges is the high heterogeneity between studies which makes it difficult to make a generalization. Moreover, confidence intervals did not include the indifference in only three cases. Furthermore, Li et al. did not report the overall FLAIR resection HR, a result that could have an important value in the final analysis. Nevertheless, despite all these elements, we obtained a slightly significant advantage on OS by performing an SMR, but it only reinforces our aforementioned considerations. In other words, the evidence to support the superiority of SMR over GTR on PFS and OS is weak.

In conclusion, except for a few studies, clear criteria for a pre-operative allocation to a more aggressive surgery do not exist. Moreover, as the patient’s post-operative neurological outcome is universally recognized as the most important factor influencing OS, it appears necessary to conduct a more detailed prospective study evaluating the anatomical, radiological, and surgical factors favoring SMR.

## 5. Conclusions

We performed a critical review of published articles concerning SMR in patients affected by glioblastoma. The definition of SMR is still debatable and no consensus exists on which radiological and clinical criteria would indicate it as a useful treatment. Similarly, which intraoperative method is the most successful in performing a safe and effective SMR has yet to be determined. Finally, the detailed analysis of post-operative neurological outcomes is limited to KPS and does not consider neurocognitive functions.

## Figures and Tables

**Figure 1 brainsci-12-00652-f001:**
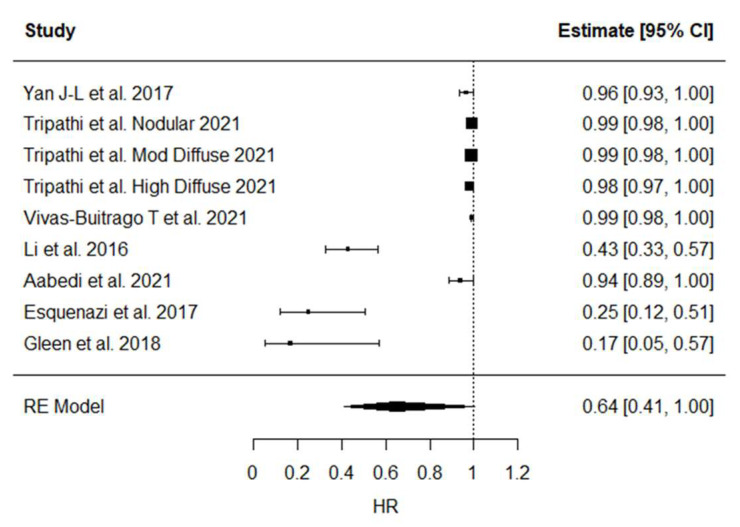
OS HRs Pooled Meta-Analysis.

**Figure 2 brainsci-12-00652-f002:**
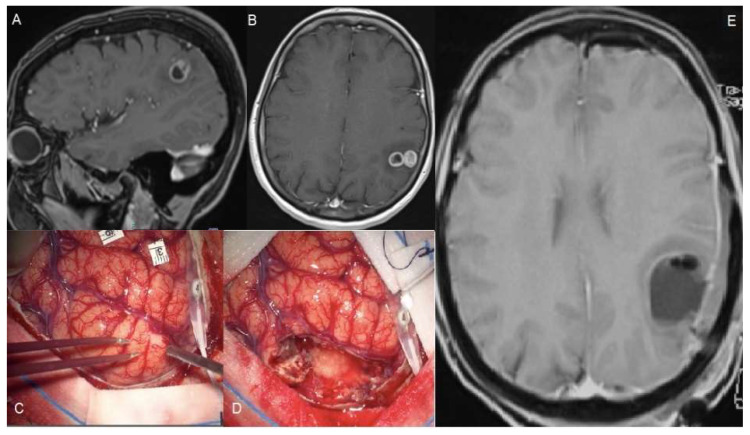
A 56-year-old female suffered a single seizure (speech articulation impairment lasting 10 min). The upper figures (**A**,**B**) show pre-operative T1 gadolinium-enhanced MRIs of a Glioblastoma infiltrating the left supramarginal gyrus. The patient was operated on through an awake craniotomy and direct language mapping (**C**,**D**). Since the mapping did not show activation areas on the supramarginal gyrus, a complete gyrus resection was performed. (**E**) The post-operative MRI confirmed the complete resection not only of the tumor but also of the gyrus. Post-operatively, the patient did not experience any speech disturbances.

**Table 1 brainsci-12-00652-t001:** *Data of clinical studies*. The table shows data collected from clinical studies. Pre T1c vol (cc): pre-operative T1w plus gadolinium tumor volume; Ependyma: involvement of ventricular of periventricular white matter; Intraop Methods: intraoperative methods; NA: not available; iMRI: intraoperative MRI; IONM: intraoperative neurophysiological monitoring; GTR: gross total resection; SMR: supramarginal resection; i-CT: intraoperative CT scan; i-US: intraoperative ultrasound.

*Authors*	*Definition of Useful SMR*	*Impact on OS*	*Pre T1c Vol (cc)*	*Location*	*Ependyma*	*Intraop Methods*	*Functional Outcomes*	*Conclusions*
*Yan J-L et al., 2017*	Resection of DTI anisotropic component (q) > 89%	≈100 days more	46 ± 30	10 in Eloquent Areas, 12 Near-Eloquent, and 9 Non-Eloquent	NA	NA	NA	DTI anisotropic q component resection is related to better PFS
*Tripathi S et al., 2021*	Resection of FLAIR volume beyond T1c: Highly diffuse: 30–99%; Moderately diffuse: 10–60%; and Nodular: 10–29%	Nearly double survival	36.2	NA	Worse Survival	NA	NA	Moderately- and highly-diffuse wtIDH gliomas benefited from SMR
*Vivas-Buitrago T et al., 2021*	20 to 50% FLAIR volume resection beyond T1c	Increased without time definition	36.2	No effect on OS	Worse Survival	NA	NA	A FLAIR resection of at least 20% but less than 60% is associated with improved OS
*Roh et al., 2020*	Frontal/temporal lobectomy on non-dominant hemisphere	≈36 months more	61.1 Frontal location and 41.9 Temporal Location	NA	NA	Tractography, neuronavigation, and 5-ALA	No difference in post-KPS	Non-dominant side GTR plus lobectomy is associated with a better OS and PFS without decreasing performance
*Li et al., 2016*	Resection of 53.21% of FLAIR beyond T1c	≈5 months more	31.0 (0.3–186.3)	NA	NA	IONM and awake surgery	More motor deficits if FLAIR EOR < 53.21%	Resection of a minimum of 53.21% of FLAIR beyond T1c is associated with improved OS
*Aabedi et al., 2021*	No advantage was found in the NCE group	None	28.8 (0.5–172.1)	37.7% Frontal, 32.0% Temporal, 20.1% Parietal, 19% Occipital, and 0.4% Insula	NA	NA	Post-operative impairment was the only factor affecting OS	Post-operative neurological impairment was the only factor affecting OS
*Certo et al., 2020*	Resection of tumoral FLAIR volume beyond T1c	NA	54.9 (33.4–89.7)	17 in Eloquent Areas, 29 Near-Eloquent, and 22 Non-Eloquent	NA	5-ALA, neuronavigation, IONM, i-CT, and i-US	No difference in post-KPS	FLAIRectomy was associated with improved OS
*Esquenazi et al., 2017*	Resection beyond T1c edges	≈37.5 months more	35.5 (0.4–107)	33% Frontal, 42% Temporal, 22% Parietal, an d3% Occipital	NA	Neuronavigation	NA	The subpial technique permitted an SMR with an improved OS, without new deficits
*Eyopoglu et al., 2016*	Resection until 5-ALA is not detectable anymore (DIVA Technique)	≈4.5 months more	30 ± 24	Advantages in Non-Eloquent and Near-Eloquent areas	NA	IONM, 5-ALA, and iMRI	No difference with control-group	DIVA technique was associated with better OS in non-eloquent and near-eloquent areas
*Gleen et al., 2018*	Resection of 1 cm beyond T1c	≈13 months more	39 (until 120)	Temporal	NA	Awake surgery	No difference between GTR and SMR groups	Temporal SMR was associated with better OS and PFS
*Mampre et al., 2018*	Resection beyond T1c edges	None	31.9 (13.9–56.1)	NA	NA	IONM and awake surgery	No significant correlation with FLAIR resection	Post-operative FLAIR volume was not associated either with PFS or OS
*Pessina et al., 2017*	Maximal safe resection of FLAIR volume	≈12.6 months more	59.1 (9.1–399.4)	20 in Eloquent areas, 198 Near-Eloquent, and 64 Non-Eloquent	NA	IONM, iUS	No difference between GTR and STR groups	A >45% resection of FLAIR volume was associated with significantly improved OS
*Shah et al., 2020*	Lobectomy in right frontal, temporal occipital, and left occipital lobes	≈16 months more	NA	59.5% Temporal, 25% Occipital, and 15.6% Frontal	NA	NA	No difference in post-KPS	Lobectomy in case of non-eloquent areas was associated with improved OS
*Altieri et al., 2019*	Altered signal intensity in FLAIR sequences	None	23.14 (0–106.56)	NA	NA	IONM and 5-ALA	NA	Resection of FLAIR areas did not affect Glioblastoma patients’ OS
*Aldave et al., 2013*	Resection until 5-ALA is not detectable anymore	≈9.5 months more	43.2	NA	NA	IONM and 5-ALA	Non-significant worse functional outcome	The absence of fluorescent residue was associated with improved OS
*Schneider et al., 2019*	Anterior temporal lobectomy	≈12 months more	≈30	Temporal	NA	5-ALA	No difference in post-KPS	Anterior temporal lobectomy was linked to lingering OS and PFS
*Figueroa et al., 2020*	Minimally invasive anterior temporal lobectomy	No difference	NA	Temporal	NA	IONM	No difference in post-KPS	Minimally invasive anterior temporal lobectomy was a feasible and safe technique

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
