# Peer review of "Supramarginal Resection for Glioblastoma: It Is Time to Set Boundaries! A Critical Review on a Hot Topic"

_brainsci, 2022, doi:10.3390/brainsci12050652_

Round 1

Reviewer 1 Report

Line 77-78 on page 2 is awkwardly written and should be rephrased.

Results section is confusing as written.  The tables are very helpful, but the text focus more on the studies which had components excluded.  It should be clarified within the results which studies and particular components were actually included.  This may need to be a separate paragraph but may help in adding to overall clarity and readability.

Overall, the authors do a fine job of displaying the message: that the evidence regarding SMR and improved outcomes are marginal at best, weak, and varied.  They highlight several important questions which such studies provide and also the heterogeneity in their methodology.  The lack of standardized definitions, outcomes, and means to measure SMR are parts of the root of the problem.  Overall it does a nice job of summarizing the current state of the literature on this issue and creates a springboard for future studies to answer these questions.  Would merely ask to address some of the comments I made above.

Author Response

Dear reviewer,

thank you for time you dedicated to the revision of our paper. Following your precious suggestions, we will try to answer to your observations.

  • “Line 77-78 on page 2 is awkwardly written and should be rephrased.”

We rewrote sentences to improve sintax.

  • “Results section is confusing as written. The tables are very helpful, but the text focus more on the studies which had components excluded.  It should be clarified within the results which studies and particular components were actually included.  This may need to be a separate paragraph but may help in adding to overall clarity and readability.”

We added an another paragraph with papers included into the review.

Finally, we are so grateful to the reviewer for final considerations and we are enjoy that message we wanted to transmit, it can be perceived by the readers.

Best regards,

Francesco Guerrini, M.D.

Reviewer 2 Report

Materials and Methods

  • The authors should list "lobectomy" associated to "gross total resection" as a keyword for database searching too, in order to find more references with the mentioned question of the paper. Known relevant paper evaluating the question of supratotal resection versus gross total resection have not been mentioned

Results

The author didnt´t mention the median KPS in the included patients by the different studies. It would be interesting and of important knowledge if any difference was made in the KPS while including the patient for supra total resection

Discussion

It´s not completely specified if the use of resection in mm or cm over the contrast enhancing has an impact or a role on OS or neurological outcome. More specific information about the difference of supramarginal resection would be important

Author Response

Dear reviewer,

thank you for time you dedicated to the revision of our paper. Following your precious suggestions, we will try to answer to your observations.

  • “Materials and Methods

The authors should list "lobectomy" associated to "gross total resection" as a keyword for database searching too, in order to find more references with the mentioned question of the paper. Known relevant paper evaluating the question of supratotal resection versus gross total resection have not been mentioned.”

We followed your suggestion and we added further papers into the review.

  • “Results

The author didnt´t mention the median KPS in the included patients by the different studies. It would be interesting and of important knowledge if any difference was made in the KPS while including the patient for supra total resection”

We added some consideration regarding KPS into the discussion. We focused on the fact that KPS does not evaluate some important cognitive aspects, such as phasia, that are important to conduct an indipendent life.

  • “Discussion

It´s not completely specified if the use of resection in mm or cm over the contrast enhancing has an impact or a role on OS or neurological outcome. More specific information about the difference of supramarginal resection would be important.”

We focused on this point into the paragraph 4.1, as we specified as in literature many dimensional cut-off exist for the definition of a “useful supramarginal resection”, leading to confusing results and difficulties in applying them during routine activity.

Hoping our answers are clear, we thank the reviewer to having had the possibility to specify some issues and finally improve our paper.

Best regards,

Francesco Guerrini, M.D.
